# Exploring Sustainable Fertilization Strategies Involving Biochar, Compost, and Inorganic Nitrogen: Impact on Nutrient Uptake, Yield, Phytochemical Accumulation, and Antioxidant Responses in Turnips

**DOI:** 10.3390/plants14040529

**Published:** 2025-02-09

**Authors:** Rui M. A. Machado, Isabel Alves-Pereira, Diogo Velez, Ana Grilo, Isabel Veríssimo, Rui Ferreira

**Affiliations:** 1MED—Mediterranean Institute for Agriculture, Environment and Development & CHANGE—Global Change and Sustainability Institute, Crop Science Department, School of Sciences and Technology, University of Évora, Pólo da Mitra, Ap. 94, 7006-554 Évora, Portugal; 2MED—Mediterranean Institute for Agriculture, Environment and Development & CHANGE—Global Change and Sustainability Institute, Chemistry and Biochemistry Department, School of Sciences and Technology, University of Évora, Colégio Luís António Verney, Ap. 94, 7006-554 Évora, Portugal; 3Chemistry and Biochemistry Department, School of Sciences and Technology, University of Évora, Colégio Luís António Verney, Ap. 94, 7006-554 Évora, Portugal; diogofrazao31@gmail.com (D.V.); sofanagrilo@gmail.com (A.G.); isabelverissimo02@gmail.com (I.V.)

**Keywords:** *Brassica rapa* L., municipal compost, carbon sequestration, circular economy, nitrate, antioxidant enzymes, soil enzymes, soil fertility

## Abstract

The effect of fertilization strategies involving biochar, compost, and inorganic N on growth and phytochemical accumulation in turnips was studied in a greenhouse pot experiment. The experiment was carried out with six fertilizer treatments: unfertilized soil (US), compost (120 g/pot) + biochar (20 g/pot) (C + B), compost (120 g/pot) + 0.5 g N/pot (C + 0.5 N), biochar (20 g/pot) + 0.5 g N/pot (B + 0.5 N), compost (120 g/pot) + biochar (20 g/pot) + 0.5 g N/pot (C + B + 0.5 N), and inorganic fertilization (pre-plant mineral fertilizer + 1 g N/pot (PF + N)). Compost, biochar, and pre-plant mineral fertilizer were incorporated into the top 10 cm of soil, while inorganic N was applied weekly in equal amounts. The application of biochar with C + 0.5 N, compared to C + 0.5 N alone, reduced plant Ca and Mg uptake but did not affect dry biomass production. Using compost + 0.5 N, with or without biochar, proved to be a viable and sustainable strategy, achieving high dry biomass yields similar to inorganic fertilization but with lower inorganic N and no addition of the other inorganic nutrients. The biomass increase was due to enhanced nutrient uptake, resulting from the synergy between compost and the application of a reduced amount of N. The co-applications reduced nitrate levels in both shoots and roots and left the soil more fertile than soil fertilized with inorganic fertilization. Fertilization strategies differently affected the accumulation of ascorbic acid, total phenols, polyphenol oxidase, and proline in roots and shoots. Inorganic fertilization enhanced ascorbic acid and total phenols in shoots, while the combination of compost, biochar, and 0.5 N was more effective in roots, suggesting that fertilization strategies should be tailored to tissue-specific goals.

## 1. Introduction

Soil degradation is a pressing global concern, highlighting the need for strategies to enhance or maintain soil fertility. Factors such as erosion, compaction, loss of organic matter, and rising salinity and acidity contribute to this decline. Alarmingly, up to 40% of arable soils are acidic [1], and approximately 20% of irrigated land is affected by salinization [2]. Soil degradation can be accelerated by climate change, driven by greenhouse gas emissions. The use of inorganic fertilizers, while essential for achieving high crop yields, carries a significant carbon footprint, substantially contributing to greenhouse gas emissions. Therefore, it is necessary to prioritize practices that reduce the carbon footprint of inorganic fertilization and contribute to maintaining or enhancing soil fertility.

The use of biochar and organic compost may help reduce the need for inorganic fertilizers and enhance soil fertility. Adding biochar to soil is seen as a promising way to improve fertility and plant growth while boosting carbon sequestration and potentially reducing nitrate leaching and nitrous oxide emissions [3,4,5]. However, despite the multiple advantages associated with biochar application to soil, its effects on crop production are often inconsistent. The application of biochar may enhance or at least not harm plant productivity [6], but it can also negatively impact crop yield [7,8,9,10]. Since the individual effects of biochar and compost have been explored, the novelty of this work lies in its integrated approach of combining biochar with compost and a reduced amount of nitrogen to optimize nutrient uptake, crop yield, and phytochemical accumulation in turnips grown in low-fertility soil. Biochar addition to soil alone has limited impacts on increasing yields. For biochar to have positive effects, it is necessary to combine its application with inorganic fertilizers [11], particularly N [12] or compost [13]. Biochar-amended soils may need additional N after biochar addition to maximize crop production [14]. Biochar may decrease soil N availability due to N immobilization [15], ammonium retention, and physical entrapment of nitrate [16], potentially limiting crop growth in the short term. However, in slightly alkaline soil, combining nitrogen fertilizer with biochar enhanced nitrogen use efficiency and corn production because nitrogen fertilizer reduced soil pH [17].

Combining biochar with compost application has been reported as a promising strategy to promote plant growth [13,18,19]. On the other hand, adding compost to acidic soils enhances their physical and chemical properties and can reduce greenhouse gas emissions (GHGs) [20]. One promising source of compost is municipal solid waste, which is becoming increasingly available in European countries due to the expansion of selective biowaste collection systems. This growing availability makes municipal solid waste compost a practical option for developing sustainable fertilization strategies. The EU Member States will be required to collect biowaste separately in the coming years [21].

The application of biochar plus compost [22] along with inorganic fertilization increased plant productivity [19]. However, the effect of the mix of biochar with organic materials on plant growth was mainly dependent on the chemical quality of the organic materials. The synergistic effects were prevalent when nitrogen-rich and lignin-poor materials were mixed with biochar [23]. According to Amlinger et al. [24], only about 5% to 15% of the total N supplied by mature compost is available in the first year after application. Therefore, the contribution of compost to soil N availability is relatively low. Consequently, the application of biochar alongside compost could also benefit from the addition of some inorganic N. On the one hand, the combined application of inorganic nitrogen with organic compost increased yield compared to the sole application of compost [25]. Additionally, applying compost and some inorganic N contributed to a reduction in the need for inorganic N [25] and other inorganic nutrients (e.g., K and P) without compromising yield [26]. Thus, we hypothesize that the combined application of biochar, compost, and inorganic nitrogen may enhance nutrient uptake, yield, and phytochemical accumulation in turnip. Thus, the main goal of this research was to evaluate the effects of fertilizing strategies that involve biochar, compost, and inorganic N on nutrient uptake, yield, root and shoot nitrate content, phytochemical accumulation, and antioxidant responses in turnips grown in low-fertility soil.

## 2. Results

### 2.1. pH, EC, and Extractable Nutrients of Compost and Biochar

The pH values of the compost (7.8) and biochar (7.1) indicate that both amendments may contribute to mitigating soil acidity. The compost had a significantly higher EC (4.41 dS m^−1^) compared to biochar (0.2 dS m^−1^). The nutrient release into the aqueous extract per gram of amendment (Table 1), which reflects the ability of amendments to supply extractable nutrients, was significantly affected by the type of amendment. The compost’s extractable K content was high (73.8 mg/g of compost), while biochar exhibited much lower levels (0.27 mg/g of biochar). In addition to K, compost can provide nitrate (1.22 mg/g), phosphate (1.46 mg/g), calcium (2.65 mg/g), magnesium (0.3 mg/g), and sulfate (0.3 mg/g) (Table 1). Extractable NO_3_^−^ from biochar was higher (5.19 mg/g biochar) than in compost. Extractable Ca^2+^ from biochar was also significant. Extractable phosphate and sulfate levels in biochar were negligible.

### 2.2. Soil pH and Enzymes

The soil pH after the harvest of turnips was significantly affected by fertilizer treatments. Soil pH in treatments with compost was higher than that in treatments without compost (Figure 1). The soil pH of plants grown with C + B, C + 0.5 N, and C + B + 0.5 N increased to values above 7.0, reaching 7.8, 7.4, and 7.2, respectively (Figure 1). The application of B + 0.5 N to the soil led to a lower pH (5.74) than C + 0.5 N (7.4). In treatments where compost was not applied, but N as ammonium nitrate was applied (B + 0.5 N and PF + N), the soil pH values decreased to below 6.

The highest alkaline phosphatase (ALP) activity was observed in the unfertilized soil (5.39 nmol·min^−1^·mg^−1^) (Figure 2a). ALP activity in fertilized treatments ranged from 2.8 to 3.3 nmol·min^−1^·mg^−1^ and was not significantly affected by the fertilizer treatment (Figure 2a).

Soils from treatments without organic amendments (US and PF + N) showed the lowest catalase activity (Figure 2b). The addition of amendments with or without N increased catalase activity. The co-applications with biochar increased catalase activity significantly relative to the treatment C + 0.5 N. Soils of plants grown with C + B + 0.5 N resulted in the highest observed catalase activity (1516.6 nmol·min^−1^/mg) (Figure 2b). The CTT activity in treatment C + B + 0.5 N was 2.34 times higher than C + 0.5 N and 3.96 times higher than PF + N.

### 2.3. Shoot and Root Nutrient Uptake

All fertilizer treatments increased the uptake of N, P, K, Ca, and Mg by roots and shoots compared to unfertilized soil (US), except for the uptake of Ca and Mg by roots in the C + B treatment (Table 2). Plants grown with C + 0.5 N took up higher root and shoot macronutrients than those cultivated with B + 0.5 N (Table 2).

The addition of biochar with C + 0.5 N compared with C + 0.5 N reduced the uptake of Ca and Mg by the shoots and P and Mg by the roots. However, it significantly increased Ca uptake by the roots (Table 2).

N uptake by roots and shoots was significantly higher in plants grown with inorganic fertilization (PF + N) (1288.1 mg/plant), exceeding N uptake in plants grown with C + B + 0.5 N by 25.3% and with C + 0.5 N by 31.3%. However, C + B + 0.5 N and C + 0.5 N led to an increase in N uptake by 105.6% and 96%, respectively, relative to inorganic N applied. Plants grown under C + 0.5 N and C + B + 0.5 N took up more K in both shoots and roots compared to those cultivated with inorganic fertilization (Table 2). These treatments increased K uptake in shoots by 122.7%. and 102.6%, respectively, and in roots by 80.1%. and 95.8%.

Plant P uptake (roots + shoots) was slightly higher in plants grown with C + 0.5 N than in those with C + B + 0.5 N and inorganic fertilization, by 7.1% and 8.1%, respectively.

Thus, despite the addition of biochar with compost + 0.5 N leading to a decrease in P uptake, its absorption was equal to that of plants grown with inorganic fertilization. Plant Ca uptake in plants grown with C + 0.5 N (227.8 mg/plant) was higher than in those grown with PF + N (189.1 mg/plant) and C + B + 0.5 N (159.6 mg/plant).

The co-application of biochar with C + 0.5 N relative to C + 0.5 N reduced the plant Ca uptake by 29.9%. Regarding Mg, plant uptake in plants grown with C + 0.5 N was similar to PF + N but 46.6% higher than in C + B + 0.5 N. The co-application of biochar with C + 0.5 N relative to C + 0.5 N reduced the plant (shoots + roots) Mg uptake by 46.6%.

Micronutrient uptake in both shoots and roots of turnips was also higher across all fertilizer treatments than in unfertilized soil (Table 3). Plants grown with C + 0.5 N took up more Fe, Mn, and Zn in their shoots, despite the higher soil pH (7.4) compared to B + 0.5 N (5.6). However, the uptake of Fe, Cu, and Mn by the roots of plants grown with B + 0.5 N was higher than in those grown with C + 0.5 N, with Fe uptake reaching 8.37 mg/plant (Table 3). Inorganic fertilization generally increased micronutrient uptake in both shoots and roots compared to C + B + 0.5 N, except for B uptake in roots (Table 3). Despite differences in nutrient uptake, plants across the treatments did not exhibit visual symptoms of nutrient deficiency or toxicity. The noticeable differences were limited to significant variations in the size of the roots and shoots.

### 2.4. Fresh and Dry Biomass Yield

All fertilizer treatments significantly increased the fresh and dry weights of both roots and shoots compared to the unfertilized soil (US) (Table 4). The application of compost and biochar, although it increased plant dry biomass, resulted in only a slight increase (128.6%) compared to C + 0.5 N (771.4%) and C + B + 0.5 N (690.5%), highlighting the importance of some N application (Table 4). The application of biochar with compost and 0.5 N did not affect the dry biomass of either the shoots or roots relative to the application of C + 0.5 N (Table 4).

The dry biomass of plants grown with compost and 0.5N, with or without biochar, and without the addition of other inorganic nutrients besides N (0.5N), was not significantly different from that of plants grown with inorganic fertilization alone (PF + N) (Table 4). The treatments C + 0.5 N, PF + N, and C + B + 0.5 N led to plant dry weight gains of 771.4%, 738.1%, and 690.5%, respectively, compared to the unfertilized soil (Table 4).

### 2.5. Nitrates

Both the roots and shoots of plants grown in unfertilized soil did not contain nitrate (Figure 2). Shoots and roots nitrate content in plants grown with amendments and 0.5N increased, ranging from 0.08 to 0.27 mg/g FW (Figure 3). However, these values were lower than those typically considered usual for turnips (0.5 to 1 mg NO_3_^−^/g FW), which are a moderately nitrate-accumulating species [27]. The co-application of biochar with C + 0.5 N decreased nitrate content in roots relative to the co-application of C + 0.5 N alone.

### 2.6. Phytochemical Accumulation

Despite all fertilization treatments increasing dry weight biomass, only the treatments C + 0.5 N and C + B + 0.5 N had biomass comparable to that obtained with inorganic fertilization. Therefore, the effects of treatments on phytochemical accumulation analysis primarily focus on these treatments. These fertilizer treatments significantly affected the content of ascorbic acid (ASA), total phenolic compounds (TPC), proline (Pro) content, and polyphenol oxidase (PPO) activity (Table 5). The addition of biochar to C + 0.5 N compared to C + 0.5 N alone increased AsA and TPC in roots and proline in both shoots and roots (Table 5). Biochar addition to C + 0.5 N relatively with C + 0.5 N and PF + N decreased the PPO activity. Among the treatments C + 0.5 N, C + B + 0.5 N, and PF + N, the highest shoot AsA content was observed in PF + N (8.4 mg/100 g FW). Shoot ascorbic acid content in plants grown with C + 0.5 N (6.0 mg/100 g FW) and C + B + 0.5 N (6.4 mg/100 g FW). was similar (Table 5). The TPC and proline levels were higher in shoots than in roots across all treatments (Table 5). However, the levels of AsA and PPO activity varied between roots and shoots depending on the fertilizer treatment. For the treatments C + 0.5 N, C + B + 0.5 N, and PF + N, the results indicate that PF + N showed the best results for AsA, TPC, and Pro in the shoot, while C + B + 0.5 N was superior for these parameters in the root and minimized PPO activity. C + 0.5 N had intermediate performance.

### 2.7. Glutathione and Ascorbate-Glutathione Cycle Enzymes

Fertilizer treatments significantly affected the enzymes of the glutathione-ascorbate cycle in both the roots and shoots of turnips (Table 6). The shoot GSH content was significantly higher in plants grown with C + 0.5 N (1.3 mg/100 g FW) compared to those grown with C + B + 0.5 N (1.0 mg/100 g FW) and PF + N (0.7 mg/100 g FW), with no significant differences between the latter two treatments. In contrast, root GSH content showed no significant differences among the C + 0.5 N, C + B + 0.5 N, and PF + N treatments. In contrast, root GSH content did not differ significantly among the treatments.

In shoots, glucose-6-phosphate dehydrogenase (G6PD) activity was lowest in the treatment where inorganic fertilization was applied (Table 6). In the roots, G6PD activity was highest in the C + 0.5 N treatment (2.6 nmol.min^−1^/mg), followed by PF + N (2.3 nmol.min^−1^/mg), with the lowest value observed in C + B + 0.5 N (2.1 nmol.min^−1^/mg).

Shoot glutathione reductase (GR) activity in the treatments C + 0.5 N, C + B + 0.5 N, and PF + N was not significantly different (Table 6). Root GR activity was higher in plants grown with C + 0.5 N (58.5 nmol.min^−1^/mg) compared to those grown with C + B + 0.5 N (17.8 nmol.min^−1^/mg) and PF + N (24.8 nmol.min^−1^/mg), although no significant differences were observed between the latter two treatments.

Plants grown with C + 0.5 N and C + B + 0.5 N had no significant differences in shoot glutathione peroxidase (GPx) activity (Table 6). However, shoot GPx in C + B + 0.5 N was higher than in PF + N. Root GPx activity was lower in plants grown with C + 0.5 N (5.4 nmol.min^−1^/mg) than in C + B + 0.5 N (14.6 nmol.min^−1^/mg) and PF + N (11.6 nmol.min^−1^/mg).

Shoot ascorbate peroxidase (APx) activity of the C + 0.5 N, C + B + 0.5 N, and PF + N treatments was not significantly different. However, among these treatments, the root APx was higher in C + B + 0.5 N (7.8 nmol.min^−1^/mg) than in PF + N (3.4 nmol.min^−1^/mg) and C + 0.5 N (2.3 nmol.min^−1^/mg) (Table 6).

The fertilizer treatments had distinct effects on shoot and root glutathione-ascorbate cycle enzymes, highlighting the different antioxidant responses in these plant tissues. For example, C + B + 0.5 N enhanced root APx activity without affecting shoot APx activity (Table 6). Similarly, C + 0.5 N decreased the GPx activity in the roots and maintained high levels of this catalytic activity in the shoots.

### 2.8. Antioxidant Activity

The effect of the treatments on the parameters used to assess antioxidant activity varied. In the treatments C + 0.5 N, C + B + 0.5 N, and PF + N, the DPPH-2,2-Diphenyl-1-picrylhydrazyl radical scavenging activity in both shoots and roots, FRAP (Ferric Reducing Antioxidant Power) in shoots, and peroxidases (POX) and catalase (CTT) were not significantly affected by the treatments (Table 7). However, root FRAP activity and root POX enzyme activity were higher in plants grown with PF + N. The results suggest that preplant fertilization with N (PF + N) enhances root antioxidant activity, while C + B + 0.5 N offers a more balanced effect, and C + 0.5 N is less effective overall.

### 2.9. Oxidative Stress and Cell Damage Markers

Fertilizer treatments significantly affected shoot and root ROS and MDA (Figure 4). The addition of biochar to C + 0.5 N reduced ROS levels by 44.2% in shoots and by 47.7% in roots, indicating a reduction in oxidative stress compared to the C + 0.5 N treatment. ROS content in shoots and roots of the C + B + 0.5 N and inorganic fertilization treatments was not significantly different (Figure 4a). Among the treatments C + 0.5 N, C + B + 0.5 N, and PF + N, the MDA levels were only affected significantly in the shoots (Figure 4b). Shoot MDA in plants grown with C + B + 0.5 N (4.13 µmol/100 g FW) was significantly higher than those grown with C + 0.5 N (2.37 µmol/100 g FW) and PF + N (2.70 µmol/100 g FW), which did not differ significantly from each other (Figure 4b).

## 3. Discussion

Contrary to the hypothesis, the application of biochar to compost and 0.5 N did not enhance macronutrient uptake compared to the co-application of compost and 0.5 N. This could be explained by the low EC value of biochar and the low levels of extractable nutrients in biochar. The low EC value of biochar (0.21 dS/m) compared with the compost (4.41 dS/m) indicates that its contribution to plant nutrition may be reduced (Table 1). The levels of extractable nutrients in biochar were very low, except for the unexpectedly high nitrate content in the aqueous extract (5.19 mg/g) (Table 1), which contrasts with reports of low extractable N in biochar [28]. However, Liu et al. [29] reported that biochar can be a potential N source for plants as it contains both organic forms of N and inorganic forms (NH_4_^+^; NO_3_^−^, and N_2_O). On the other hand, extractable K was relatively low, which diverges from the typically high extractable K levels in biochar reported by Prasad et al. [30]. The differences likely reflect biochar’s inherent variability, influenced by factors such as feedstock type, pyrolysis temperature, particle size, and residence time, as well as differences in nutrient extraction methodologies. The methodology used in this study differed from the one that is normally used to determine extractable nutrients. This difference could have influenced the determined values.

The addition of biochar to compost and 0.5 N, unexpectedly, had no significant effect on pH and, consequently, may have had a limited impact on plant nutrition. This could be explained by the lower application rate of biochar or its limited capacity to increase pH. The alkalinity of biochar varies depending on the feedstock and pyrolysis conditions [31].

The addition of biochar with C + 0.5 N appeared to alter the uptake by tissues, reducing the uptake of Ca and Mg by shoots and P and Mg by roots while significantly increasing Ca uptake by the roots. This suggests that biochar could influence nutrient distribution within the plant, potentially altering nutrient transport and/or availability in the soil. Further research is necessary to understand why biochar reduces plant Ca uptake while increasing its accumulation in the roots.

The higher N uptake by roots and shoots subjected to inorganic fertilization was due to the higher amount of N applied, which was more than double the amount used in treatments with 0.5 N. Despite this, the N levels in plants grown with C + 0.5 N and C + B + 0.5 N may have reached the sufficiency range, owing to the synergy between compost and reduced N application, which enhanced plant N uptake, increasing it by approximately 1.85-fold compared to inorganic N application (500 mg).

Plant K uptake was higher in plants grown in C + 0.5 N and C + B + 0.5 N treatments than in those grown with inorganic fertilization only. These co-applications likely enhanced K uptake by stimulating plant growth through improved N availability, which increased K absorption from the soil, where its availability was high due to high extractable K provided by compost and an increase in pH. Compost can improve plant nutrition even before organic matter mineralization, particularly through its significant contribution of extractable potassium (73.8 mg/g) alongside nitrate, phosphate, calcium, magnesium, and sulfate (Table 1). The elevated K levels in the aqueous extract of compost can be attributed to the water-soluble nature of K in organic matter, which facilitates its rapid release into the soil solution [32,33]. Unlike N and P, which are often bound to organic compounds and require microbial activity for release, K is not strongly associated with organic matter decomposition. This characteristic enables compost to supply K to plants almost immediately after application. The addition of organic fertilizer increased K availability in the soil at all stages of maize growth. and it increases with the volume of irrigation water applied [34].

In plants grown with fertilization, the increase in plant P uptake relative to unfertilized soil ranged from 167.1% to 860%. This is consistent with the elevated ALP activity in the unfertilized soil, which was likely due to the limited availability of inorganic P. Phosphatase activity is one of several adaptive mechanisms in the plant–soil system to counteract P deficiency, facilitating the mineralization of organic P to increase its availability for plants and soil organisms [35]. In contrast, the lower ALP activity in fertilized soils indicates a higher availability of inorganic P, which reduces the need for enzymatic mineralization of organic P. The highest plant P uptake in plants grown with C + 0.5 N can be attributed to the increased soil pH, which enhances P availability, as well as the greater phosphate availability provided by the compost. Furthermore, the humic substances in compost can enhance diffusive efflux, which can improve P availability for crops [26]. Additionally, the combination of organic and mineral fertilizers has been shown to improve phosphorus solubilization and reduce P precipitation, further enhancing its availability [36]. The co-application of biochar with C + 0.5 N reduces P uptake slightly (294.1 mg/plant), but it did not differ significantly from inorganic fertilization (291.4 mg/plant). Thus, the co-application of compost and 0.5 N with or without biochar may reduce or even eliminate the need for inorganic K and P. Machado et al. [26] also reported that the combined application of compost with N can replace inorganic P and K fertilization. Machado et al. [26] also reported that the combined application of compost with N can replace inorganic P and K fertilization.

Plants grown with C + 0.5 N alone had higher Ca and Mg uptake than those treated with PF + N, highlighting the effectiveness of the co-application of compost and some N in providing these nutrients. The addition of biochar to C + 0.5 N reduced plant Ca and Mg uptake, with a 29.9% decrease in Ca and a 46.6% reduction in Mg compared to C + 0.5 N alone.

Regarding micronutrients, the addition of biochar to compost and 0.5N improved shoot and root B and Cu uptake compared to C + 0.5N, but had lower Mn and Zn uptake in both shoots and roots (Table 3). Inorganic fertilization increased micronutrient uptake compared to C + B + 0.5 N, probably due to low pH (Figure 1). In treatments where compost was not applied but ammonium nitrate was used (B + 0.5 N and PF + N), the reduction in soil pH to values below 6.0 can be attributed to the acidifying effect of ammonium. Ammonium nitrate is an acidifying fertilizer, as ammonium releases H^+^; ions into the soil during the nitrification process [37]. On the other hand, when roots uptake ammonium, it releases H+ into the soil solution, decreasing rhizosphere pH [38]. Although turnips thrive in soils with a slightly acidic to neutral pH, ideally ranging from 6.0 to 7.0 [39], differences in soil pH across treatments can influence plant nutrient uptake by affecting nutrient availability. Soil pH decreases generally increase the bioavailability of Fe, Mn, B, Zn, and Cu [40]. However, the exception of B uptake suggests that further research is needed to better understand how compost and biochar influence micronutrient uptake. Despite the differences in nutrient uptake, no visual symptoms of deficiency or toxicity were observed. The only noticeable differences were the significant variations in the size of the roots and shoots.

Contrary to what was initially hypothesized, the combined application of biochar, compost, and inorganic N did not enhance dry biomass yield but enhanced soil fertility, as evidenced by the highest increased soil catalase activity observed after the turnip harvest (Figure 2b). Geng et al. [41] and Yao et al. [42] have also reported that biochar application increases catalase activity in soil. The increased catalase activity, a marker of enhanced microbial activity and the soil’s ability to mitigate oxidative stress, combined with a near-neutral soil pH (Figure 1), supports the conclusion that the C + B + 0.5 N treatment contributed more significantly to improving soil fertility than the other treatments [43]. Since the dry biomass of the treatments C + 0.5 N, C + B + 0.5 N, and inorganic fertilization were similar, this indicates that using compost with 0.5 N, either without or with biochar, may reduce or even eliminate the need for inorganic fertilization with K, P, Ca, and Mg. Thus, the C + 0.5 N with or without biochar emerges as the most sustainable fertilization option because it balances high dry biomass productivity with a reduction in the inputs of inorganic fertilizers and resource reutilization. Moreover, the nitrate content in both shoots and roots of plants grown with C + 0.5 N and C + B + 0.5 N was lower than that of plants grown with inorganic fertilization. Vieira et al. [44] also reported that leaf nitrate content increased linearly with the amount of N applied. N fertilization facilitates the accumulation of nitrate in plant tissues as a result of an excess of N uptake over its reduction [45]. In plants grown with inorganic fertilization, the high amount of N applied likely facilitated nitrate accumulation in plant tissues due to an excess of N uptake over its reduction [45].

Nitrate concentrations in the roots and shoots of plants grown with organic amendments and 0.5 N were similar across treatments. However, in plants grown with inorganic fertilization, nitrate concentrations were higher in the shoots than in the roots (Figure 2). This contrasts with the findings of Antonious et al. [46], who reported higher nitrate content in roots than in shoots.

Considering the previous results, the discussion on the effects of fertilizers on phytochemicals, enzymes of the glutathione-ascorbate cycle, antioxidant activities, oxidative stress, and cell damage markers will focus exclusively on the influence of the C + 0.5N, C + B + 0.5 N, and inorganic fertilization treatments. Once again, the co-application of biochar with compost and 0.5 N did not consistently enhance the accumulation of all the phytochemicals evaluated. The treatments influence phytochemical accumulation differently in the roots and shoots, highlighting their potential for targeted fertilization strategies. Inorganic fertilization enhances AsA levels in shoots, while the C + B + 0.5 N treatment should be used to maximize AsA levels in roots. The addition of biochar to C + 0.5 N compared to C + 0.5 N alone increased AsA, TPC in roots, and Pro in both shoots and roots (Table 7). According to Mishra et al., Chowdhary et al., Kumar et al., and Petcu et al. [47,48,49,50], such increases in antioxidants enhance stress tolerance and improve cellular protection. Biochar addition also reduced PPO activity, which is advantageous as high PPO activity is linked to reduced shelf life due to post-harvest deterioration [51]. This suggests that the C + B + 0.5 N treatment may extend the shelf life of turnips by minimizing post-harvest losses. The higher TPC and proline levels observed in shoots compared to roots suggest differential allocation of secondary metabolites. Yang et al. [52] also reported that total phenol content was higher in leaves than in roots.

The fertilizer treatments had distinct effects on the enzymes of the glutathione-ascorbate cycle, demonstrating tissue-specific antioxidant responses. Thus, in the C + 0.5 N treatment, a higher shoot GSH content suggests an enhanced capacity for redox regulation and antioxidant defense in above-ground tissues. Additionally, this treatment exhibited the highest G6PD activity and higher GR activity in roots, which may enhance NADPH production, a critical coenzyme for GR activity, thereby promoting GSH regeneration, a tripeptide that plays a vital role in ROS scavenging and redox homeostasis maintenance by the glutathione-ascorbate cycle [53,54,55]. In the C + B + 0.5 N treatment, the enhanced root GPx and APx activities indicate that adding biochar to C + 0.5 N promotes efficient hydrogen peroxide scavenging in roots. These effects may improve the plant’s oxidative stress resilience compared to inorganic fertilization [56,57]. In contrast, the PF + N treatment exhibited lower G6PD activity, suggesting a reduced capacity for ROS scavenging compared to organic amendments combined with 0.5 N.

The increased root FRAP and POX activities in plants grown with inorganic fertilization (Table 7) suggest that inorganic fertilization may enhance the activity of heme-containing oxidoreductases. by facilitating the oxidation of a broad range of organic and inorganic electron donor substrates through reactions with hydrogen per-oxide or organic hydroperoxides, thereby mitigating the oxidative stress caused by heavy metals in the roots [58]. The POX activity values are of the same order of magnitude as those determined by Lalay et al. [59].

Once again, the results do not highlight a standout treatment, as the treatments affected ROS and MDA levels differently in turnip tissues. Elevated ROS levels observed in plants treated with C + 0.5 N (Figure 4) highlight an imbalance likely caused by the metabolic conditions under this treatment, possibly due to differences in nutrient uptake or distribution. ROS are commonly recognized as byproducts of stress-related metabolic processes [60,61]. Interestingly, while the C + B + 0.5 N treatment reduced ROS levels, it resulted in significantly higher MDA levels in shoots compared to C + 0.5 N and PF + N. This indicates that lipid peroxidation was more pronounced in the C + B + 0.5 N treatment, possibly due to localized oxidative damage in shoot tissues despite reduced ROS levels. The lower MDA levels in the PF + N treatment suggest that inorganic fertilization was effective in maintaining lipid membrane integrity in shoots. Despite differences in ROS and MDA levels, no significant effects were observed on the dry biomass of turnip roots and shoots, nor were there visible signs of chlorosis or necrosis, indicating that the plants tolerated oxidative stress without any noticeable impact on growth. However, further studies are required to better understand the effects of these fertilization strategies on plant resilience under abiotic stress.

## 4. Materials and Methods

### 4.1. Growth Conditions and Amendments

The study was conducted in a greenhouse located at the “Herdade Experimental da Mitra” (38°57′ N, 8°32′ W), University of Évora, Portugal. The greenhouse was covered with polycarbonate and had no supplemental lighting. Radiation data inside the greenhouse were not available. Air temperatures inside the greenhouse ranged from 3 to 26 °C, and outside solar radiation ranged from 31.3 to 226.9 W·m^−2^·d^−1^.

The experiment was carried out in plastic pots. Each 12 L plastic pot (21 cm high × 27 cm diameter) was filled with 14 kg of loamy sandy soil obtained from the upper 30 cm of soil of the Mitra Research Farm, Portugal. The soil presented a 1.2% organic matter content, a bulk density of 1.47 g·cm^−1^, and a pH of 5.7 (1:2.5 soil-to-water distilled water ratio, *w*/*v*), an electrical conductivity (ECe) of a saturated paste extract of 0.3 dS m^−1^, 75 mg K·kg^−1^, 79 mg P·kg^−1^, 1.16 meq Ca^2+^/100 g, and 0.57 meq Mg^2+^/100 g.

The experiment was carried out with six fertilization treatments, which involved the application of compost (120 g/pot), biochar (20 g/pot), inorganic fertilizers applied in pre-plantation, and inorganic N applied weekly. The characteristics of compost and biochar, along with their raw material origins, are detailed in [26,62]. Biochar was pyrolyzed at a temperature of 400 to 500 °C and was used 11 months after its production.

The treatments were as follows: unfertilized soil (US), compost + biochar (C + B). compost + 0.5 g N/pot (C + 0.5 N), biochar + 0.5 g N/pot (B + 0.5 N), compost + biochar + 0.5 g N/pot (C + B + 0.5 N), and inorganic fertilization, consisting of pre-plant mineral fertilizer plus 1 g inorganic N per pot (PF + N) (Table 8). The inorganic nitrogen doses of 0.5 g N/pot and 1 g N/pot were supplied as ammonium nitrate (16.9% NO_3_^−^–N and 16.7% NH_4_^+^–N) and applied as weekly topdressings in equal amounts, starting at transplantation and finishing two weeks before harvest.

Fifteen days prior to transplanting, mature municipal solid waste organic compost in pulverulent form, biochar, and inorganic fertilizers applied in pre-plantation were added to each pot of respective treatment and incorporated into the top 10 cm of soil. In pre-plantation, the inorganic nutrients applied were 0.17 g N, 0.36 g P_2_O_5_, 0.59 g K_2_O, 0.33 g CaO, and 0.13 g MgO. Treatments were arranged in a randomized complete block design with five replicate pots per treatment.

Turnip seedlings (*Brassica rapa* L. cv. Falko) were transplanted on 7 January 2023, into 12 L pots. Plant irrigation was based on volumetric soil water content measurements taken daily (8:00–9:00 h) with a soil-moisture probe (SM105T Delta devices, Cambridge, UK). When the average soil moisture in the top 0.1 m, measured at 7 cm from the center of the pots, of treatment PF + 1 N was ≤20%, plants were watered by hand (9:00–10:00), avoiding applying high volumes of water to minimize drainage losses. The weeds were regularly removed from the pots manually.

The plants were harvested 62 days after transplantation and separated into shoots and roots. Samples of these were stored at −80 °C for chemical and biochemical analyses.

### 4.2. Measurements

#### 4.2.1. Determination of pH, EC, and Extractable Nutrients of Compost and Biochar

The pH, electrical conductivity (EC), and concentrations of NO_3_^−^, Ca^2^^+^; PO_4_^3−^, K^+^; Mg^2^^+^; and SO_4_^2−^ in the aqueous extract (1:5, amendment to water, *w*/*v*) of the compost and biochar were measured. The pH and EC were determined using a potentiometer (pH Micro 2000, Crison, Barcelona, Spain) and a conductivity meter (LF 330 WTW, Weinheim, Germany), respectively.

The concentrations of NO_3_^−^, Ca^2^^+^; PO_4_^3−^, K^+^; Mg^2^^+^; and SO_4_^2−^ were measured using the HI93728, HI937521, HI93706, HI93750, HI937520, and HI93751 kits, respectively, in a HI 83225 Nutrient Analysis photometer (Hanna Instruments, Woonsocket, RI, USA).

#### 4.2.2. Dry Biomass, Nutrient and Nitrates Content

Four plant shoots and roots from each treatment were washed and oven-dried at 70 °C for 2–3 days, weighed, and ground so that they would pass through a 40-mesh sieve. Then analyzed for N, P, K, Ca, Mg, and nitrates. N was analyzed by using a combustion analyzer (Leco Corp., St. Josef, MI, USA). The K was analyzed by flame photometry (Jenway, Dunmow, UK). The P was analyzed using a UV/Vis spectrometer (Perkin Elmer Lamba 25). The Ca and Mg were analyzed using an atomic absorption spectrometer (Perkin Elmer, Inc., Shelton, CT, USA).

Shoots and roots NO_3_^−^ determination was realized in accordance with methodology described by Lastra [63,64].

### 4.3. Biochemical Characterization of Soil and Plants

#### 4.3.1. Soil Extracts and pH Determination

After harvesting the plants, three soil cores were randomly collected from each pot using a soil probe with a diameter of 3 cm and a depth of 10 cm. The samples were then mixed to form a composite sample, which was used to determine soil pH and to prepare extracts for measuring soil enzyme activities (alkaline phosphatase and catalase). Soil pH was measured in 1:2.5 soil-to-water suspensions (*v*/*w*) using a potentiometer (pH Micro 2000, Crison).

To prepare the buffered soil extract (SB), 10 mL of 50 mM acetate buffer (pH 5.0) was added to 1 g of soil. The mixture was shaken at 250 rpm for 1 h to facilitate extraction, followed by centrifugation at 4000× *g* for 5 min. The resulting supernatant was carefully collected and stored at −20 °C for subsequent determination of enzyme activities, following the methodology described by Fornasier et al. [65].

#### 4.3.2. Plant Extracts

The methanolic extract (MW) was prepared following the method described by Lichtenthaler and Buschmann [66]. Briefly, 1 g of shoot or root material from each pot was macerated in a mortar and homogenized with 8 mL of an 80:20 (*v*/*v*) methanol:water solution for 1 min. The homogenate was then centrifuged at 6440× *g* for 5 min at 4 °C. The resulting extracts were stored in aliquots at −20 °C for subsequent determination of phytochemical content and antioxidant activity.

The phosphate buffer extract (PB) was prepared by macerating 1 g of shoot or root samples in liquid nitrogen (Air Liquide, Lisboa, Portugal) and homogenizing them in 5 mL of 0.12 mM phosphate buffer at pH 7.2. The supernatant, obtained by centrifuging the extract at 15,000× *g* for 15 min at 4 °C, was collected and stored in aliquots at −20 °C for subsequent analysis of enzyme activities and stress markers [67,68].

#### 4.3.3. Phytochemical and Antioxidant Activity Determination

The total phenolic content (TPC), ascorbic acid (AsA) content, DPPH—2,2-diphenyl-1-picrylhydrazyl antioxidant activity, and FRAP—ferric reducing antioxidant power of turnip shoots and roots were determined in the MW, following the methodology described by Machado et al. [62].

Proline was measured in the MW [69], while glutathione (GSH) and protein contents were assessed in the PB using the methods of Machado et al. [26] and Lowry et al. [70], respectively.

Reactive oxygen species (ROS) levels in shoots and roots were determined in the PB following LeBel’s method [71]. This method relies on the reaction of 2′,7′-dichlorofluorescein (DCFH) with ROS to produce the fluorescent compound DCF, which was quantified using fluorescence measurements with an excitation wavelength of 488 nm and an emission wavelength of 525 nm at 25 °C.

Malondialdehyde (MDA) was assessed as an index of lipid peroxidation based on the quantification of TBA oxidation products. Fluorescence measurements were taken at an excitation wavelength of 515 nm and an emission wavelength of 553 nm at 25 °C, using MDA generated from 1,1,3,3-tetramethoxypropane through acid hydrolysis as the standard [72]. MDA was determined in the PBE extract of shoots and roots.

All spectrometric measurements were performed using a Hitachi U-2001 double-beam spectrophotometer (Hitachi, Ltd., Tokyo, Japan) with temperature regulated by a Grant water circulation bath (Grant Instruments, Ltd., Cambridge, UK).

All fluorometric measurements were performed in a Shimadzu RF-5001PC single-beam spectrofluorophotometer (Shimadzu Corpotation, Kyoto, Japan).

#### 4.3.4. Enzyme Activities of Soils and Plants

Soil alkaline phosphatase activity (ALP, EC 3.1.3.1) was determined in the SB by monitoring the hydrolysis of the chromogenic substrate p-nitrophenyl phosphate (pNPP) and measuring the absorbance increase at 405 nm due to p-nitrophenol (pNP) formation. Aliquots of the extract, at an appropriate protein concentration, were incubated in molecular absorption cells containing the reaction mixture of 1.2 mM pNPP in 0.05 M Tris-HCl buffer (pH 8.5). ALP activity was calculated from the slope of linear reaction curves over 180 s at 37 °C, using a molar absorptivity coefficient of 16.03 M^−1^ cm^−1^ for pNP [73].

Catalase activity (CTT, EC 1.11.1.6) was determined by measuring the decrease in absorbance at 240 nm due to H_2_O_2_ consumption, following [74]. The reaction mixture contained 30 mM H_2_O_2_ and the appropriate concentration of soil SB or shoot/root PB in 50 mM phosphate buffer (pH 7.5). Catalase activity was calculated using the slope of the linear reaction curves and a molar absorptivity coefficient of 0.0435 mM^−1^·cm^−1^ for H_2_O_2_.

Polyphenol oxidase activity (PPO, EC 1.14.18.1) was determined by measuring absorbance at 420 nm using a reaction mixture of 0.1 M catechol in 0.1 M sodium acetate buffer (pH 6) and shoot or root PB. The blank contained 0.2 M phosphate buffer (pH 6.5). Absorbance was recorded for 300 s at 37 °C, and enzyme activity was calculated from the reaction curve slope using a molar absorptivity coefficient of 1150 M^−1^·cm^−1^ for o-quinone [75].

Glucose-6-P dehydrogenase activity (G6PD, EC 1.1.1.49) was measured as described by Postma et al. [76] in a reaction mixture containing 400 μM NADP^+^, 5 mM MgCl_2_·6H_2_O, and an appropriate concentration of shoot or root PB in 50 mM Tris-HCl (pH 8). The reaction was initiated by adding 5 mM glucose-6-P, and NADPH formation was monitored by absorbance at 340 nm for 180 s. G6PD activity was calculated using the molar absorptivity coefficient for NADPH (6.22 mM^−1^·cm^−1^).

Glutathione reductase (GR, EC 1.6.4.2) and peroxidase (POX, EC 1.11.1.7) activities were measured using an appropriate concentration of PB, following the method described by Machado et al. [77].

Glutathione peroxidase (GPx, EC 1.11.1.9) and ascorbate peroxidase (APx, EC 1.11.1.11) activities were determined using the appropriate concentration of PB, in accordance with the method outlined by Machado et al. [62].

All enzyme measurements were performed using a Hitachi U-2001 double-beam spectrophotometer (Hitachi, Ltd., Tokyo, Japan), with temperature controlled by a Grant water circulation bath (Grant Instruments, Ltd., Cambridge, UK).

### 4.4. Data Analysis

Data were processed via analysis of variance using SPSS Statistics 29 software (Chicago, IL, USA). licensed to the University of Évora. Means were separated at the 5% level using Duncan’s new multiple range test.

## 5. Conclusions

The combined application of biochar with compost and a reduced amount of inorganic nitrogen did not enhance nutrient uptake or the dry biomass yield of turnip shoots or roots compared to the application of compost and a reduced amount of inorganic nitrogen alone. However, post-harvest, soils treated with the biochar combination remained more fertile, indicating potential long-term benefits for soil health.

## Figures and Tables

**Figure 1 plants-14-00529-f001:**
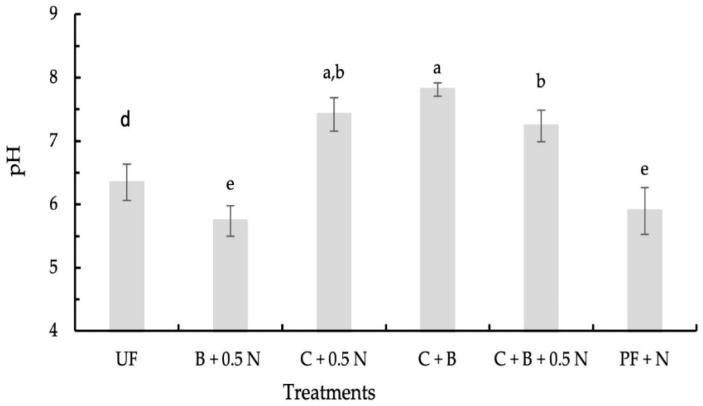
Effect of fertilizer treatments on soil pH after turnip harvest in the upper 10 cm of soil (US—unfertilized soil. C—compost. B—biochar. N—nitrogen. PF—pre-plant fertilization). Means with different letters are significantly different at *p* < 0.05. Each bar represents the mean of five replicates, and the error bars represents ±1 SE.

**Figure 2 plants-14-00529-f002:**
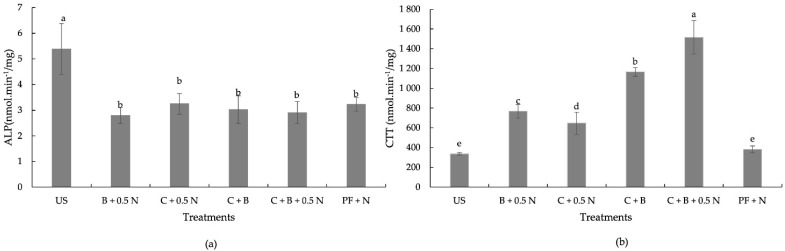
Effect of fertilization treatments on soil enzyme activities ALP (**a**) and CTT (**b**) in the 0–10 cm soil layer after turnip harvest (US—unfertilized soil. C—compost. B—biochar. N—nitrogen. PF—pre-plant fertilization. ALP—alkaline phosphatase activity. CTT—Catalase activity). Means with different letters are significantly different at *p* < 0.05. Each bar represents the mean of five replicates, and the error bars represents ±1 SE.

**Figure 3 plants-14-00529-f003:**
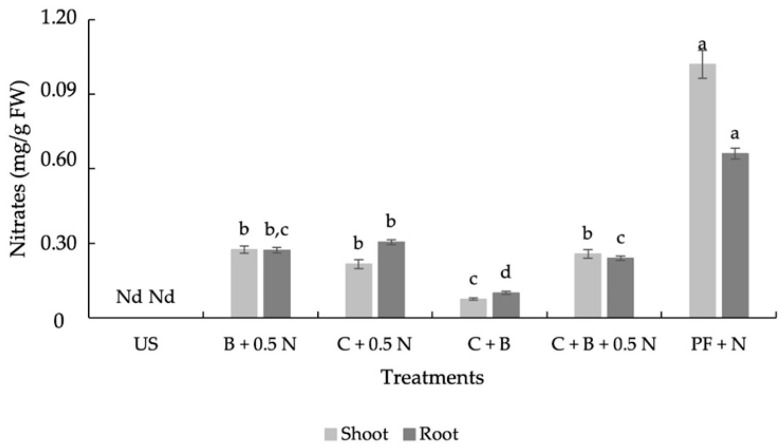
Effect of fertilization treatments on shoot and root nitrate content (US—unfertilized soil. C—compost. B—Biochar. N—nitrogen. PF—pre-plant fertilization, FW—fresh weight, Nd—not detected). Means with different letters are significantly different at *p* < 0.05. Each bar represents the mean of five replicates, and the error bars represents ±1 SE.

**Figure 4 plants-14-00529-f004:**
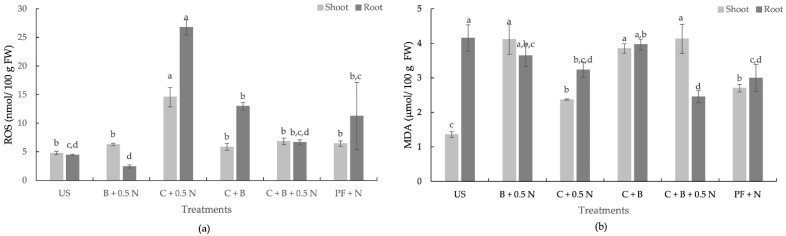
Effect of fertilization treatments on shoot and root ROS and MDA of turnip (US—unfertilized soil. C—compost. B—Biochar. N—nitrogen. PF—pre-plant fertilization, ROS—reactive oxygen species (**a**). MDA—malondialdehyde) (**b**); FW—fresh weight. Means with different letters are significantly different at *p* < 0.05. Each bar represents the mean of five replicates, and the error bars represent ±1 SE.

**Table 1 plants-14-00529-t001:** pH, EC, and the extractable nutrients per gram of amendment.

	pH	Ec	NO_3_^−^	K^2+^	PO_4_^3−^	SO_4_^2−^	Ca^2+^	Mg^2+^
Amendment		dS m^−1^	(mg/g Amendment)					
Compost	7.8 ^1^	4.41	1.22	73.8	1.46	0.3	2.65	0.3
Biochar	7.1	0.21	5.19	0.27	0.0	0.0	2.35	0.05
Significance	***	***	***	***	***	***	***	***

^1^ Each value represents the mean of four replicates. *** significant at *p* < 0.001.

**Table 2 plants-14-00529-t002:** Effect of fertilizer treatments on N, P, K, Ca, and Mg uptake by turnip shoots and roots.

	Shoot Macronutrients (mg/Shoot)					Root Macronutrients (mg/Root)				
Treatments	N	P	K	Ca	Mg	N	P	K	Ca	Mg
US	54.7 e ^1^	12.1 e	44.1 f	32.8 e	5.5 e	28.6 e	20.7 e	35.9 f	10.5 d	4.5 d
B + 0.5 N	412.4 c	52.3 c	116.7 e	137.3 b	33.4 b	294.7 d	93.0 c	165.0 d	26.2 c	36.7 b
C + 0.5 N	532.3 b	91.9 b	505.1 a	203.3 a	32.5 b	448.5 c	223.0 a	519.6 b	24.5 c	53.9 a
C + B	125.8 d	29.2 d	166.2 e	69.6 d	17.2 d	70.9 d	58.4 d	129.3 e	10.1 d	6.6 d
C+ B + 0.5N	552.4 b	96.4 b	459.8 a	107.5 c	28.7 c	475.7 b	197.7 a,b	564.8 a	52.1 a	17.4 c
PF + N	760.7 a	108.8 a	226.9 b	157.5 b	55.0 a	527.4 a	182.6 b	288.5 c	31.6 b	28.6 b
Significance	***	***	***	***	***	***	***	***	***	***

^1^—Means followed by different letters within a column are significantly different. *** significant at *p* < 0.001 (US—unfertilized soil. C—compost. B—biochar. N—nitrogen. PF—pre-plant fertilization).

**Table 3 plants-14-00529-t003:** Effect of fertilizer treatments on Fe, B, Cu, Mn, and Zn uptake by turnip shoots and roots.

	Shoot Micronutrients (mg/Shoot)					Root Micronutrients (mg/Root)				
Treatments	Fe	B	Cu	Mn	Zn	Fe	B	Cu	Mn	Zn
US	0.6 d ^1^	0.02 d	0.18 d	0.20 e	0.11 d	0.71 e	0.04 d	0.21 e	0.17 f	0.18 c
B + 0.5 N	2.9 b	0.13 c	0.84 b	0.66 c	0.63 b	8.37 a	0.15 c	2.64 a	1.96 a,b	0.65 b
C + 0.5 N	3.4 a	0.09 d	0.99 b	1.10 a,b	0.81 a	2.97 d	0.15 c	1.20 c	1.12 d	0.84 b
C + B	1.1 c	0.04 d	0.34 c	0.33 d	0.11 d	0.95 e	0.07 d	0.52 d	0.38 e	0.16 c
C+ B + 0.5N	3.8 a	0.18 c	1.39 a	0.94 d	0.46 c	3.59 c	0.32 a	1.56 b	1.03 d	0.63 b
PF + N	3.8 a	0.22 a	1.29 a	1.51 a	0.75 a	6.40 b	0.26 b	2.14 a	1.80 b	0.94 a
Significance	***	***	***	***	***	***	***	*	*	***

^1^—Means followed by different letters within a column are significantly different. *, *** significant at, *p* < 0.05, and *p* < 0.001 levels, respectively (US—unfertilized soil. C—compost. B—biochar. N—nitrogen. PF—pre-plant fertilization).

**Table 4 plants-14-00529-t004:** Effects of fertilizer treatments on root and shoot fresh weight and dry weight of turnip.

Treatments	Shoot	Root	Plant	Shoot	Root	Plant	Plant DW Increase ^4^
	FW ^2^ (g/Plant)			DW ^3^ (g/Plant)			(%)
US	23.2 e ^1^	43.6 e	66.8 e	1.6 e	2.6 e	4.2 e	
B + 0.5 N	102.8 c	146.0 c	248.8 c	8.2 c	13.1 c	21.3 c	407.1
C + 0.5 N	160.8 b	241.6 b	402.4 b	12.1 a	24.5 a	36.6 a	771.4
C + B	49.2 d	111.2d	160.4 d	3.6 d	6.0 d	9.6 d	128.6
C + B + 0.5 N	165.4 b	252.4 b	417.8 b	11.5 a,b	21.7 a,b	33.2 a,b	690.5
PF + N	172.4 a	270.4 a	442.8 a	12.7 a	22.5 a,b	35.2 a	738.1
Significance	***	***	***	***	***	***	-

^1^—Means followed by different letters within a column are significantly different. *** significant at *p* < 0.001 (US—unfertilized soil. C—compost. B—Biochar. N—nitrogen. PF—pre-plant fertilization), ^2^—FW—fresh weight, ^3^—DW—dry weight, ^4^—Increase in plant dry weight compared to unfertilized soil.

**Table 5 plants-14-00529-t005:** Effects of fertilizer treatments on phytochemical content and turnip PPO activity.

	AsA		TPC		PPO		Proline	
	mg/100 g FW ^2^		mg GAE/100 g FW		nmol min^−1^/mg		mg/100 g FW	
Treatments	Shoot	Root	Shoot	Root	Shoot	Root	Shoot	Root
US	5.7 b,c ^1^	9.5 b	22.1 b,c	10.6 a	122.6 a	58.3 c	14.3 c	5.3 c
B + 0.5 N	4.4 c	12.8 a,b	19.1 c	7.7 b	69.3 b	62.6 c	37.1 a,b	29.9 b
C + 0.5 N	6.0 b,c	10.8 b	28.6 a,b	7.4 b	61.7 b	97.8 b	31.1 b	31.1 b
C + B	6.9 a,b	4.6 c	27.8 b	11.6 a	11.1 d	169.6 a	19.8 c	6.0 c
C + B + 0.5 N	6.4 b	15.4 a	29.1 a,b	12.0 a	23.6 d	40.1 d	37.8 a,b	41.5 a
PF + N	8.4 a	11.4 b	35.7 a	8.1 b	45.4 c	47.7 c,d	46.9 a	33.5 b
Significance	***	***	***	***	***	***	***	***

^1^—Means followed by different letters within a column are significantly different. *** significant at *p* < 0.001 (US—unfertilized soil. C—compost. B—Biochar. N—nitrogen. PF—pre-plant fertilization. AsA—ascorbic acid. TPC—total phenol compounds. PPO—Polyphenol oxidase enzyme activity), ^2^—FW fresh weight.

**Table 6 plants-14-00529-t006:** Effects of fertilizer treatments on turnip root and shoot glutathione-ascorbate cycle.

	GSH		G6PD		GR		GPx		APx	
	mg/100 g FW ^2^		nmol.min^−1^/mg							
Treatments	Shoot	Root	Shoot	Root	Shoot	Root	Shoot	Root	Shoot	Root
US	1.0 b ^1^	1.7 a,b	3.9 a	1.5 d	19.2 c	24.0 c	58.6 b,c	28.3 a	1.1 c	1.9 d
B + 0.5 N	1.1 a,b	2.0 a	2.2 c	1.0 e	42.3 a	68.8 a	84.5 a	2.1 c	1.5 a	1.6 d
C + 0.5 N	1.3 a	1.9 a	2.3 b,c	2.6 a	25.3 b	58.5 b	53.2 b,c	5.4 c	1.2 b,c	2.3 d
C + B	0.9 b,c	1.3 b	1.9 d	2.5 a	28.1 b	15.3 c	55.9 b,c	14.0 b	1.2 b,c	11.3 a
C + B + 0.5 N	1.0 b,c	1.6 a,b	2.5 b	2.1 c	28.0 b	17.8 c	63.5 b	14.6 b	1.4 a,b	7.8 b
PF + N	0.7 c	2.1 a	1.6 e	2.3 b	22.8 b,c	24.8 c	46.4 c	11.6 b	1.3 a,b,c	3.4 c
Significance	***	***	***	***	***	***	***	***	***	***

^1^—Means followed by different letters within a column are significantly different. *** significant at *p* < 0.001 (US—unfertilized soil. C—compost. B—biochar. N—nitrogen. PF—pre-plat fertilization. GSH—glutathione. G6PD—Glucose-6-Phosphate Dehydrogenase, GR—glutathione reductase, GPx—glutathione peroxidase. APx—ascorbate peroxidase). ^2^—FW fresh weight.

**Table 7 plants-14-00529-t007:** Effects of fertilizer treatments on turnip root and shoot antioxidant activity.

	DPPH		FRAP		POX		CTT	
	mgGAE/100 g FW ^2^		Mg Trolox E/g FW		nmol.min^−1^/mg		μmol.min^−1^/mg	
Treatments	Shoot	Root	Shoot	Root	Shoot	Root	Shoot	Root
US	28.1 a ^1^	11.1 b	197.4 a	37.8 c	12.1 c	348.4 a,b,c	329.5 c	410.4 b
B + 0.5 N	28.6 a	10.6 b	33.8 b	263.3 b	49.1 a,b	330.2 b,c	479.7 b	136.5 d
C + 0.5 N	28.3 a	12.1 a,b	22.9 b	248.2 b	17.7 b,c	392.3 a,b	409.4 b,c	81.2 e
C + B	28.8 a	14.3 a	38.3 b	211.5 b	45.3 a	194.4 d	696.3 a	632.0 a
C + B + 0.5 N	28.2 a	12.5 a,b	39.4 b	233.0 b	33.8 a,b,c	249.9 c,d	335.0 c	162.8 d
PF + N	34.7 a	10.3 b	34.9 b	318.8 a	19.1 a,b,c	456.0 a	403.6 b,c	317.1 c
Significance	NS	***	***	***	***	***	***	***

^1^—Means followed by different letters within a column are significantly different. *** significant at *p* < 0.001. NS = not significant, (US—unfertilized soil. C—compost. B—Biochar. N—nitrogen. PF—pre-plant fertilization. DPPH—2,2-Diphenyl-1-picrylhydrazylradical scavenging activity. FRAP—Ferric Reducing Antioxidant Power Assay. POX—peroxidase activity. CTT—catalase activity), ^2^—FW fresh weight.

**Table 8 plants-14-00529-t008:** Designation used for the six fertilization treatments that constitute the experiment.

Designation	Description
US	unfertilized soil
C + B	compost + biochar
C + 0.5 N	compost + 0.5 g N/pot
B + 0.5 N	biochar + 0.5 g N/pot
C + B + 0.5 N	compost + biochar + 0.5 g N/pot
PF + N	pre-plant mineral fertilizer + 1 g inorganic N/pot

C—Compost (120 g/pot), B—biochar (20 g/pot), 0.5 N—0.5 g inorganic N/pot, N—1 g inorganic N/pot.

## Data Availability

The data is contained in the article.

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
