# Peer review of "Exploring Sustainable Fertilization Strategies Involving Biochar, Compost, and Inorganic Nitrogen: Impact on Nutrient Uptake, Yield, Phytochemical Accumulation, and Antioxidant Responses in Turnips"

_plants, 2025, doi:10.3390/plants14040529_

Round 1

Reviewer 1 Report

Comments and Suggestions for Authors

Detailed comments are attached

Reviewer 2 Report

Comments and Suggestions for Authors

The submitted manuscript to PLANTS-MDPI entitled “Exploring Sustainable Fertilization Strategies Involving Biochar, Compost, and Inorganic Nitrogen: Impact on Nutrient Uptake, Yield, Phytochemical Accumulation, and Antioxidant Responses in Turnips” is interesting to investigate. But the following are the comments that need to be addressed before acceptance:

What was the novelty of this work since extensive research has already been done on the sustainable fertilization strategies?

Line 22: The amount of compost and biochar should be mentioned in the previous sentance.

Line 25-26: What was the rationale behind increasing dry biomass after compost application?

Line 29: Name the phytochemicals.

There are some very small paragraphs in the introduction which should be logically combined.

Line 84-86: There are three grammatical mistakes in a single sentence i.e. “Enhances, increases, optimizes”

The results and discussion sections should be separated.

Table 2 & 9: Overlapped by the line numberings.

There should be standard deviation in the tables.

Results: Rather than mentioning the values (which can also be seen from the figures and tables), please mention the change in a percentage or fold-change.

I Strongly recommend authors to replace the tables with some high-quality figures. 2-3 tables are enough in a single manuscript.

The mechanistic diagram that elucidates the key findings and conclusions of this study should be added.

Line 480: Why was there no lightning?

A table should be added in the M*M section consisting of the treatment’s explanation.

The conclusion should be completed in a single paragraph.

After the inclusion of the above-mentioned suggestions, this manuscript can be accepted for publication in PLANTS.

Round 2

Reviewer 1 Report

Comments and Suggestions for Authors

I suggest that the authors can combine and simplify some indicators in the "discussion" section according to the objectives and main conclusions of the paper, to better increase the integrity of the paper.

Author Response

Response to reviewer

We agree with the reviewer's suggestion and appreciate their valuable feedback. Accordingly, we have revised the discussion section, with the modified text highlighted in green.